# Confronting pastoralists' knowledge of cattle breeds raised in the extensive production systems of Benin with multivariate analyses of morphological traits

**Sandrine O. Houessou**[1], **Luc Hippolyte Dossa** [1]*, **Rodrigue Vivien Cao Diogo**[2], **Maurice Cossi Ahozonlin**[1], **Mahamadou Dahouda**[1], **Eva Schlecht**[3]

**1** Ecole des Sciences et Techniques de Production Animale, Faculté des Sciences Agronomiques, Université d'Abomey- Calavi, Bénin, **2** Département des Sciences et Techniques de Productions Animale et Halieutique, Université de Parakou, Faculté d'Agronomie, Bénin, **3** Animal Husbandry in the Tropics and Subtropics, University of Kassel and University of Göttingen, Germany

☯ These authors contributed equally to this work.
* hippolyte.dossa@fsa.uac.bj

**Data Availability Statement:** All relevant data are within the manuscript and its Supporting Information files.

## Abstract

Cross-border seasonal livestock movements in West Africa bring into close contact several cattle breeds. In the coastal countries hosting migrating herders from the Sahel, it often affects the genetic variability and geographical distribution of traditional cattle breeds, through their indiscriminate but also intended crossbreeding with larger-framed Sahelian cattle breeds. The need to secure and effectively manage this genetic variability, in order to respond to changing production and market conditions, is widely recognized by the scientific community, livestock herders and policy-makers. This however requires a comprehensive knowledge of the breeds' characteristics. The indigenous criteria used by pastoralists to characterize and distinguish cattle breeds remain unclear and further validation is required. This study was therefore designed to document and validate herders' knowledge on cattle breeds. From June 2015 to June 2016, 803 cattle herders participated in a phenotypic breed description in seven pastoral communities across the country. Each cattle herder was asked to name and describe morphologically the different cattle breeds in his herd. Subsequently, fifteen body measurements taken on a total of 1401 adult cattle (964 cows and 439 bulls) were submitted to multivariate analyses. Participants distinguished ten different cattle breeds kept in traditional herds according to six primary morphological traits and clearly separated zebuine from taurine breeds. These results were consistent with those of the multivariate analyses of the measured traits. However, herders' classification approach proved to be more accurate in distinguishing breeds within the zebuine subspecies. Hence, while metric measurements and molecular genetic analyses are promising approaches to fill the knowledge gap on the diversity of local farm animal genetic resources, they should integrate livestock herders' traditional knowledge for more precision.

**Funding:** This work was supported by Volkswagen Foundation, Germany (DE) to LHD, Grant number: Az 89 367, URL: https://www.volkswagenstiftung. de/. The funders had no role in study design, data collection and analysis, decision to publish, or preparation of the manuscript.

## Introduction

West Africa is rich in a wide variety of cattle breeds represented by the taurine (*Bos taurus*) and Zebu subspecies [1,2]. Exposed to complex social, political and environmental processes over centuries, these genetic resources have developed under harsh conditions to which they have adapted [3]. They are appreciated for their adaptive traits which include the resistance to diseases and drought, ability to walk long distances, and capacity to survive on poor pastures [2]. Yet, this valuable diversity is increasingly threatened by genetic dilution due to changes in production systems, livestock herders' preferences for specific breeds and/or traits, market conditions and opportunities [4]. Several studies revealed admixtures among the taurine and zebu subspecies [5,6,7] as the result of the continuous genetic flow that occurs every year during seasonal cross-border livestock movements from the drier Sahelian zones in the north to the more humid zones in the south of West and East Africa. The importance of these local genetic resources for the livelihoods of livestock herders and the sustainability of the production systems in which they are raised calls for the urgent need to promote their sustainable use [2,8,9,10] and conservation [11]. The latter author has argued that biodiversity conservation and food security are two sides of the same coin.

Yet, as revealed by the review of existing literature, the first obstacle to sustainable management of local farm animal genetic resource in African livestock production systems is the insufficient knowledge on their specific features and genetic diversity [2,8]. In Benin, like in several African countries, there are virtually no inventories of these resources and thus no reliable data available. The precise identification of animal types and breeds, and an improved understanding of their values or adaptive traits are thus necessary but depend on the availability of accurate and comprehensive information on their characteristics as well as their production and marketing environments.

The Second Report on the State of the World's Animal Genetic Resources for Food and Agriculture [12] acknowledged recent and ongoing efforts of description and characterization of livestock breeds in several West African countries. But most of these efforts tend to be fragmentary and limited either to their phenotypic [13,14,15,16], or genotypic and molecular characterization [7,17,18,19,20,21] out of the production system context, paying little attention to the local knowledge of the communities who keep them. Given its importance, livestock herders' indigenous knowledge has been recommended to be an integral part of breed characterization [22]. This knowledge is mainly useful in quantitative morphological characterization that represents the first step in the characterization process [23,24,25] and can provide, to some extent, a reasonable representation of genetic difference among populations [26]. This is more evident in Sub-Saharan Africa, where, cattle are basic assets of cattle herders who mostly are still involved in pastoralism, which is still the dominant ruminant livestock system [27]. Hence, cattle breeds are subject of much discussion among these herders [28] who have accumulated a wealth of untapped knowledge of these farm animal genetic resources and of their production environments [29,30,31,32].

Multivariate discriminant analyses of morphological traits have been reported, in several previous studies, to be effective for a precise and objective discrimination of different populations of cattle [13,14,15,33]; goats [26,34], sheep [35,36,37], and horses [38]. Therefore, results of such analyses may represent an objective basis for comparison with herders' indigenous knowledge. So far, however, no studies of which we are aware combine the two approaches in order to test their complementarity and validate pastoralists' classification of cattle breeds. This study aimed to document and validate herders' knowledge of differences among cattle breeds raised in Benin with quantitative data.

## Material and methods

### Study sites

The study was conducted in seven (07) localities which are representative of three vegetation zones along the geographical north-south gradient in Benin (Fig 1). Ferralitic soils and a bimodal rainfall with an average annual precipitation of 1250 mm characterize the regions of Kétou and Agonli in the Guinea-Congolian zone (GCZ). In the Guinean zone (GSZ), which includes Savalou and Tchaourou, the soils are of ferruginous type and the annual precipitation averages 1150 mm with a bimodal and unimodal rainfall pattern in Savalou and Tchaourou respectively. The Sudanian zone (SZ) included Kandi, Sinendé and Boukombé. It is characterized by ferruginous soils and a bimodal rainfall pattern with an average annual precipitation of 1000 mm. These locations, whose some demographic characteristics are presented in Table 1, were selected because of the presence of relatively high numbers of resident cattle herds but also because of the important influx of seasonal and cyclical migrating herds in search of pasture and water. These migrating herds are further designed as transhumant herds.

## Data collection

### Ethical statement

The study involved taking body measurements from cattle with the consent and in the presence of the cattle herder. There is no specific legislation for body measurements and hence no approval was necessary. All the data was collected in traditional farms and the animal owners agreed to be involved in the project through the Communal Sector for Agricultural Development (SCDA) which is the decentralized institution for the management of the agricultural sector in the surveyed municipalities. All the animals included in this study were managed in accordance with the criteria for the assessment of animal welfare identified by the Welfare Quality Project (WQP) [42].

### Assessment of pastoralists' knowledge about cattle breeds' characteristics

First, informal interviews were organized with all actors involved in livestock production, including officers of local extension services in each of the research locations, to obtain some preliminary information about cattle production, lists of cattle camps and of the places where cattle herds and herders gather.

Individual interviews were then conducted from June 2015 to June 2016 with 803 resident cattle herders randomly selected from those who had at least 10 years of experience in cattle herding and had spent at least five (05) years in the locality. Unequal distribution of the livestock camps resulted in unbalanced samples across the studied villages (Table 1). The herders were asked to name in their local language the breeds of cattle kept in their area and to indicate the most relevant traits they used to distinguish between them. Breed was defined according to [43] as: "either a sub-specific group of domestic livestock with definable and identifiable external characteristics that enable it to be separated by visual appraisal from other similarly defined groups within the same species, or a group for which geographical and/or cultural separation from phenotypically similar groups has led to acceptance of its separate identity". Thus, cattle breed in this study is referred to cattle of similar physical features perceived by herders as being of the same genetic origin.

### Morphological characterization

In each surveyed herd, one mature cow and, where available, one breeding bull representative of each cattle breed present and named by the herder were then randomly selected for

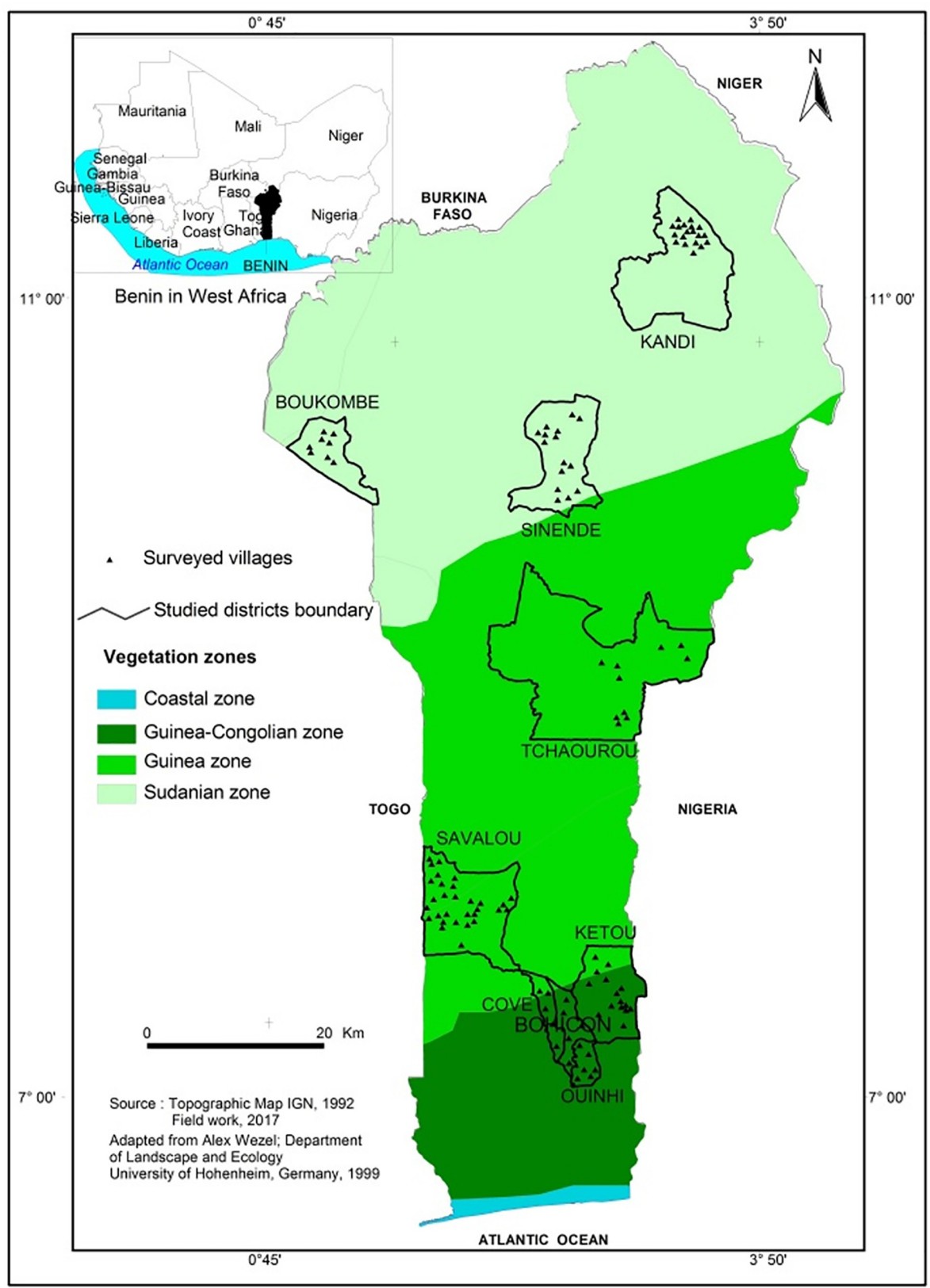

**Fig 1. Map of Benin showing the locations of the municipalities investigated.**

**Table 1. General characteristics of the study locations and number of cattle herds surveyed in Benin.**

| Location | Geographic coordinates | Climate [39] | Area (km$^2$) | Population density (people/km$^2$) [40] | Estimates of the total cattle population (heads) [41] | Cattle herds sampled (n) |
|---|---|---|---|---|---|---|
| **Kétou** | 7° 21′ N 2° 36′ E | Sub-equatorial | 1775 | 88 | 16000 | 120 |
| **Agonli** | 7° 13′ N 2° 20′ E | Sub-equatorial | 1758 | 132 | 6170 | 50 |
| **Tchaourou** | 8° 53′ N 2° 36′ E | Tropical subhumid | 7256 | 30 | 47000 | 110 |
| **Savalou** | 7° 55′ N 1° 58′ E | Tropical subhumid | 2674 | 54 | 32000 | 120 |
| **Kandi** | 11° 07′N 2° 56′ E | Dry tropical | 3421 | 52 | 159000 | 148 |
| **Sinendé** | 10° 20′ N 2° 22′ E | Dry tropical | 2289 | 39 | 80000 | 135 |
| **Boukombé** | 10° 11′ N 1° 06′ E | Dry tropical | 1 036 | 80 | 30100 | 120 |

morphometric measurements. They maturity was ascertained by the visual examination of their dentition (possession of either three or four pairs of permanent teeth). A total of 1401 adult cattle of both sexes (943 cows and 458 bulls) was measured and included in this study.

Fifteen body measurements and ten qualitative traits (Table 2) were assessed following the FAO guidelines for phenotypic characterization of animal genetic resources [25]. The qualitative traits described were sex, presence of horn, horn color, horn shape, cephalic profile, ear shape, ear orientation, body coat color and body coat color pattern, presence of hump and dewlap size. The body measurements were carried out using a measuring stick, a measuring tape or wooden caliper on animals standing on a level surface and maintained in upright posture by their respective owners.

## Statistical analysis

Collected data were processed and analyzed using SAS software, version 9.2 (SAS Institute, Inc., Cary, NC, USA). Metric data were first checked for consistency. This resulted in the exclusion of the five animals named by herders as of *Djelliji* breed and one male of *Dageeji* breed, which are not included in the 1401 individual cattle.

Descriptive analyses were firstly performed to explore statistical differences among the different cattle breeds. Frequencies and chi-square (χ2) tests for independence were performed on the aforementioned qualitative traits to explore statistical differences among breeds. Similarly, the quantitative variables were subjected to analysis of variance using PROC GLM. The least square means (LSMEANS) were calculated for males and females separately, and for both together. Comparison of means between groups was performed using the Students Newman and Keuls (SNK) multiple mean comparison tests. The results were used to screen for the most useful variables for further discriminant analysis. The canonical discriminant analysis using the CANDISC procedure was performed to determine the best linear combination of the quantitative variables that would group or separate the named cattle breeds. Canonical variables that summarize between-breeds variation were generated and the pairwise squared Mahalanobis distances calculated. PROC GPLOT was used to plot the individuals onto canonical variables for visual examination of the ordering of the different breeds in the multivariate space. The ability of these canonical functions to assign each individual animal to its original group was calculated as the percentage of correct assignment to each genetic group using the DISCRIM procedure (Nearest Neighbour Discriminant Analysis). The degree of morphological similarity or divergence between the investigated cattle breeds was assessed and their classification in homogenous groups was carried out using the method of Ascending Hierarchical Clustering (AHC) according to the criterion of the jump of Ward in the PROC CLUSTER and

**Table 2. Morphological traits measured on 1401 individual cattle across seven locations in Benin.**

| Variable | Description |
|---|---|
| **Quantitative** (in cm) | |
| Height at withers (WH) | Vertical distance from the bottom of the front foot to the highest point of the shoulder between the withers |
| Rump height (RH) | Distance from the highest point of rump to the ground |
| Heart girth (HG) | Circumference of body just behind the forelegs |
| Body length (BL) | Distance between the horn site to tail drop |
| Scapula-ischial length (SIL) | Distance from tip of the shoulder to the ischial tuberosity |
| Face length (FAL) | Distance from between the horn site to the lower lip |
| Ear length (EL) | Distance from the point of attachment to the tip of the ear |
| Head width (HW) | Distance between the most prominent points of the zygomatic arches |
| Tail length (TL) | Distance from the tail drop to the tip of the tail |
| Hip Width (HW) | Distance between the ends of the bone of the iliac crest |
| Horn length (HL) | Distance from the root of the horn to its tip along the outer curvature |
| Hock circumference (HC) | Circumference taken just above the hock joint |
| Muzzle circumference (MC) | Complete distance around the outside of the mouth |
| Chest depth (CD) | Vertical distance from the apex of the withers to the bottom of the chest |
| Shoulder point width (SPW) | Distance between the right and left shoulder points |
| **Qualitative** | |
| Sex | Male, female |
| General aspect of the coat | Uniform, Spotted, Composed |
| Unique color of coat | Black, White, Dark red, Brown, Fawn |
| Other color of coat | White spotted black, Black spotted white, White spotted red, Red spotted white |
| Cephalic profile | Concave, Convex, Straight |
| Presence of hump | Absent, Present |
| Presence of horn | Absent, Present |
| Color of horn | Black, Brown, White, Black and Brown, Black and White, Brown and white |
| Horn shape | Straight, Crown, Cup, Folded back cup, Crescent, Lyre, Folded back lyre, Wheel, Spiral, Numeral three |
| Ear shape | Rounded, Pointed |
| Orientation of ear | Erected, Horizontal, Dropping |

TREE procedures. The Mahalanobis distances generated during the canonical discriminant analysis were used to construct a dendrogram using the Unweighted Pair Group Method Analysis (UPGMA). Finally, the association between the qualitative traits was investigated through a Multiple Correspondence Analysis (MCA) using PROC CORRESP.

# Results

## Herders' perceptions of morphological characteristics of the main cattle breeds

A total of ten cattle breeds were identified from the surveyed herders' responses and named in *Fulfulde* language, spoken by almost all of the herders, as follows: *Boboji*, Somba, *Yakanaji*, Goudali/*Bokoloji*, *Bodeeji*/Bororo, *Djelliji*, *Dageeji/Dage*, *Bargouji/Muti*, *Keteeji* and one unnamed type perceived as crossbreed.

**Table 3. Key morphological traits used by herders (%) to classify nine cattle breeds raised in Benin.**

| Trait | Trait expression | Cattle breeds | | | | | | | |
|---|---|---|---|---|---|---|---|---|---|
| | | *Yakanaji* | *Goudali* | *Bodeeji* | *Djelliji* | *Dageeji* | *Bargouji/Keteeji* | *Boboji* | *Somba* |
| **Coat color** | | (n = 407) | (n = 463) | (n = 422) | (n = 84) | (n = 261) | (n = 298) | (n = 111) | (n = 100) |
| | White | 100.0 | 84.9 | 0.0 | 0.0 | 100.0 | 95.6 | 0.0 | 0.0 |
| | Reddish Brown / Black | 0.0 | 0.0 | 100.0 | 0.0 | 0.0 | 0.0 | 0.0 | 0.0 |
| | White/Reddish | 0.0 | 0.0 | 0.0 | 98.8 | 0.0 | 0.0 | 0.0 | 0.0 |
| | Variable | 0.0 | 0.0 | 0.0 | 1.2 | 0.0 | 4.4 | 100.0 | 100.0 |
| | White and black neck | 0.0 | 15.1 | 0.0 | 0.0 | 0.0 | 0.0 | 0.0 | 0.0 |
| **Body size** | | (n = 477) | (n = 396) | (n = 376) | (n = 215) | (n = 217) | (n = 400) | (n = 215) | (n = 220) |
| | Large | 95.6 | 100.0 | 99.7 | 96.7 | 0.0 | 0.0 | 0.0 | 0.0 |
| | Medium | 4.4 | 0.0 | 0.0 | 3.3 | 100.0 | 0.0 | 0.0 | 0.0 |
| | Small | 0.0 | 0.0 | 0.3 | 0.0 | 0.0 | 100.0 | 0.0 | 0.0 |
| | Dwarf | 0.0 | 0.0 | 0.0 | 0.0 | 0.0 | 0.0 | 100.0 | 100.0 |
| **Size of horn** | | (n = 527) | (n = 557) | (n = 418) | (n = 209) | (n = 251) | (n = 380) | (n = 170) | (n = 219) |
| | Absent | 0.0 | 100.0 | 0.0 | 0.0 | 0.0 | 0.0 | 0.0 | 0.0 |
| | Long | 100.0 | 0.0 | 100.0 | 0.0 | 100.0 | 0.0 | 0.0 | 0.0 |
| | Medium | 0.0 | 0.0 | 0.0 | 100.0 | 0.0 | 0.0 | 0.0 | 0.0 |
| | Short | 0.0 | 0.0 | 0.0 | 0.0 | 0.0 | 100.0 | 100.0 | 100.0 |
| **Size of hump** | | (n = 337) | (n = 374) | (n = 242) | (n = 202) | (n = 116) | (n = 418) | (n = 138) | (n = 210) |
| | Absent | 0.0 | 0.0 | 0.0 | 0.0 | 0.0 | 35.6 | 100.0 | 100.0 |
| | Well developed | 100.0 | 100.0 | 62.8 | 100.0 | 19.0 | 0.0 | 0.0 | 0.0 |
| | Poorly developed | 0.0 | 0.0 | 37.2 | 0.0 | 81.0 | 0.0 | 0.0 | 0.0 |
| | Small | 0.0 | 0.0 | 0.0 | 0.0 | 0.0 | 64.4 | 0.0 | 0.0 |
| **Size of dewlap** | | (n = 174) | (n = 223) | (n = 163) | | | (n = 120) | | |
| | Well developed | 0.0 | 100.0 | 0.0 | - | - | 0.0 | - | - |
| | Poorly developed | 100.0 | 0.0 | 100.0 | - | - | 0.0 | - | - |
| | Small | 0.0 | 0.0 | 0.0 | - | - | 100.0 | - | - |
| **Size of sheath/ umbilical folds** | | (n = 202) | (n = 247) | (n = 54) | | (n = 44) | (n = 82) | | |
| | Well developed | 100.0 | 100.0 | 0.0 | - | 0.0 | 0.0 | - | - |
| | Poorly developed | 0.0 | 0.0 | 100.0 | - | 36.4 | 0.0 | - | - |
| | Small | 0.0 | 0.0 | 0.0 | - | 63.6 | 100.0 | - | - |

The following morphological traits were used by herders to distinguish among different cattle breeds: body size (dwarf, small, middle, large), size of horn (absent, short, middle, long, long and white), hump size (absent, small, poorly developed, well developed), size of sheath/ umbilical fold (small, poorly developed, well developed), dewlap size (small, poorly developed, well developed), coat color (white, reddish-brown/ black, white/reddish, variable, white and black neck).

Table 3 presents the characteristics of the different cattle breeds according to the participating herders. The two taurine breeds, namely Somba and *Boboji* (S1 and S2 Figs), were described with similar characteristics (variable coat color, very small size, short horns and absence of hump).

The five zebu breeds (*Yakanaji*, *Goudali*, *Bodeeji*, *Djelliji*, and *Dageeji*) (S3–S7 Figs) also shared many similarities. With the exception of the hornless *Goudali* and *Djelliji* whose horn was perceived by respondents as of medium size, the zebus were, in their majority, described as large-framed and long-horned animals with a well or poorly developed hump. The Bodeeji breed was further differentiated from the Yakanaji breed by its reddish brown to red dark coat color and white horns.

Moreover, the *Dageeji* breed, in contrast to others, was perceived as a medium-sized zebu with a poorly developed hump and a small sheath/ umbilical fold.

The *Bargouji* breed (S8 Fig) was described as a shorthorn taurine-like cattle, but with a larger body size than taurine, a white coat, absence of hump or presence of small hump, a small dewlap and a small sheath/ umbilical fold. The morphological characteristics of *Keteeji* cattle were not presented separately because herders perceived *Bargouji/Muti* and *Keteeji* as two morphologically close cattle breeds. They however assumed that the pure *Bargouji* is generally humpless or has only a small hump whereas *Keteeji* is always humped with a small or poorly developed hump. They also asserted that *Keteeji* is sometimes larger in size than *Bargouji*.

## Comparison of measured morphometric traits among breeds

The differences between the cattle breeds for the morphometric variables of male and female animals and their pooled data are presented in Table 4, Table 5 and S1 Table, respectively. All linear body measurements significantly varied (P<0.05) among breeds. Major morphological traits such as height at withers (WH), rump height (RH), body length (BL) and scapulo-ischial length (SIL) were significantly greater (P<0.05) for humped cattle breeds compared with humpless ones. Within breeds, they were also significantly greater (P<0.05) in bulls than in cows. The highest mean values of WH were recorded in *Bodeeji* cows followed by the *Yakanaji* cows whereas the lowest values were obtained in the *Boboji* and Somba cows. In terms of morphometric traits, these two taurine breeds presented no significant differences.

Within the recorded qualitative traits, only a few (presence of hump, shape and orientation of horns002C size of dewlap) were useful in discriminating the zebus from the taurine breeds (S2 and S3 Tables). The white color was dominant in the zebus except for the zebu *Bodeeji*

**Table 4. Least square means (± standard error) and pairwise comparison of morphological traits measured in cows across nine cattle breeds raised in Benin.**

| Trait | Cattle breeds | | | | | | | | |
|---|---|---|---|---|---|---|---|---|---|
| | *Bargouji* | *Boboji* | *Bodeeji* | *Dageeji* | *Goudali* | *Keteeji* | Crossbreed | Somba | *Yakanaji* |
| | (n = 206) | (n = 58) | (n = 26) | (n = 23) | (n = 22) | (n = 138) | (n = 94) | (n = 84) | (n = 312) |
| MC | 40.6$^c$ ± 0.37 | 43.1$^b$ ± 0.60 | 46.5$^a$± 0.52 | 44.0$^b$± 0.44 | 40.3$^c$± 1.07 | 46.8$^a$± 0.32 | 43.6$^b$±0.36 | 40.1$^c$± 0.37 | 44.5$^b$±0.21 |
| HW | 19.5$^b$ ± 0.15 | 19.2$^b$ ± 0.23 | 20.1$^b$±0.18 | 19.2$^b$± 0.28 | 20.4$^b$± 0.57 | 23.5$^a$±0.33 | 20.3$^b$±0.24 | 17.4$^c$± 0.16 | 20.4$^b$±0.15 |
| FAL | 45.7$^c$ ± 0.42 | 43.9$^d$ ± 0.56 | 48.4$^b$± 0.21 | 47.6$^{bc}$± 0.67 | 46.0$^c$±0.60 | 46.9$^a$± 0.47 | 46.9$^{bc}$±0.36 | 39.9$^e$± 0.41 | 46.8$^{bc}$±0.3 |
| EL | 19.5$^{cd}$ ± 0.22 | 17.4$^e$ ± 0.22 | 20.6$^c$ ± 0.26 | 19.0$^d$± 0.23 | 21.9$^b$± 0.63 | 23.4$^a$± 0.48 | 20.5$^c$±0.27 | 15.4$^f$± 0.13 | 20.7$^c$±0.18 |
| HL | 37.0$^e$ ± 0.51 | 28.1$^f$ ± 1.46 | 65.8$^a$±1.86 | 49.1$^{bc}$±1.74 | 14.1$^g$± 4.10 | 47.7$^c$ ±1.04 | 42.4$^d$±1.51 | 15.8$^g$± 1.46 | 52.8$^b$±0.68 |
| HG | 152.8$^d$±0.99 | 148.9$^d$ ± 1.49 | 179.4$^a$±1.42 | 169.1$^b$ ±2.12 | 171.8$^b$±2.60 | 159.1$^c$±0.91 | 166.4$^b$±1.15 | 142.5$^e$± 1.42 | 171.4$^b$±0.93 |
| HC | 35.6$^d$±0.25 | 39.6$^c$ ± 0.44 | 44.3$^b$± 0.60 | 47.4$^a$ ±0.93 | 40.4$^c$±0.86 | 38.5$^c$±0.26 | 44.5$^b$±0.45 | 35.4$^d$± 0.54 | 43.5$^b$±0.62 |
| TL | 99.0$^{ab}$ ± 0.83 | 87.6$^b$ ± 1.01 | 103.6$^{ab}$± 1.06 | 97.8$^{ab}$± 0.69 | 103.9$^{ab}$±2.50 | 102.7$^{ab}$±1.35 | 109.7$^a$±9.65 | 91.7$^{ab}$±0.74 | 103.1$^{ab}$±0.65 |
| SPW | 34.7$^b$ ± 0.27 | 28.2$^d$ ±0.44 | 33.3$^a$± 0.76 | 30.4$^c$± 0.42 | 33.9 ±1.03 | 36.4$^a$±0.44 | 32.3$^b$±0.42 | 29.0$^{cd}$ ±0.43 | 36.6$^b$±0.38 |
| HW | 42.7$^b$ ± 0.23 | 41.0$^c$ ±0.51 | 45.8$^a$± 0.63 | 44.3$^{ab}$±0.37 | 46.0$^a$±1.01 | 45.1$^a$±0.31 | 44.3$^{ab}$±0.35 | 36.4$^d$ ±0.37 | 46.4$^a$±0.25 |
| CD | 63.9$^a$ ±0.35 | 54.6$^c$ ±0.72 | 66.1$^a$±0.68 | 64.6$^a$±0.90 | 64.9$^a$ ±1.46 | 67.0$^a$±0.51 | 60.8$^b$±0.46 | 49.8$^d$± 0.37 | 65.7$^a$±0.51 |
| WH | 115.9$^d$ ± 0.63 | 110.5$^f$±0.83 | 137.8$^a$± 0.85 | 129.1$^b$± 1.32 | 130.3$^b$± 1.30 | 131.1$^b$±0.53 | 123.4$^c$±0.78 | 100.4$^f$±0.47 | 131.4$^b$±0.76 |
| RH | 117.4$^d$ ±0.74 | 113.1$^e$±0.85 | 136.4$^a$±0.68 | 129.0$^b$±1.02 | 129.8$^{bc}$±1.05 | 134.1$^a$±0.61 | 125.4$^c$±0.74 | 105.3$^f$±0.51 | 132.4$^b$±0.65 |
| BL | 115.9$^d$ ± 0.63 | 114.6$^d$±1.07 | 132.6$^a$±0.94 | 124.4$^{bc}$± 1.18 | 120.8$^c$± 2.44 | 129.6$^a$±0.85 | 121.7$^c$±0.74 | 100.2$^e$±0.76 | 128.2$^b$±0.66 |
| SIL | 164.2$^d$ ±0.75 | 150.4$^e$±1.74 | 193.2$^a$± 2.69 | 180.3$^{bc}$±1.49 | 173.9$^c$± 1.97 | 175.1$^c$±1.25 | 177.0$^c$± 2.22 | 141.1$^f$±0.78 | 184.4$^b$±1.39 |

$^{abcdefg}$ Means with different superscript in the same row are significantly different (P ≤ 0.001), SNK's multiple mean comparison test

MC: Muzzle circumference, HW: Head width, FAL: Face length, EL: Ear length, HL: Horn length, HG: Heart Girth, HC: Hock circumference, TL: Tail length, SPW: Shoulder point width, HW: Hip Width, CD: Chest depth, WH: Withers height, RH: Rump height, BL: Body length, SIL: Scapula-ischial length.

**Table 5. Least square means in cm (± standard error) and pairwise comparison of morphological traits measured in bulls of nine cattle breeds raised in Benin.**

| Traits | Cattle breeds | | | | | | | |
|---|---|---|---|---|---|---|---|---|
| | *Bargouji* | *Boboji* | *Bodeeji* | *Goudali* | *Keteeji* | Crossbreed | Somba | *Yakanaji* |
| | (n = 131) | (n = 5) | (n = 10) | (n = 10) | (n = 93) | (n = 16) | (n = 109) | (n = 63) |
| MC | $43,5^b$±0,31 | $44,8^{abc}$±1,96 | $44,2^{abc}$±0,94 | $45,6^{abc}$±1,09 | $46,2^a$±0,39 | $47,5^a$±1,19 | $42,2^b$±0,34 | $48,0^a$±0,52 |
| HW | $20,1^{ab}$±0,17 | $19,8^{ab}$±0,20 | $21,6^{ab}$±0,58 | $22,0^{ab}$±1,01 | $21,8^{ab}$±0,27 | $23,1^a$±1,02 | $18,0^b$±0,17 | $22,5^a$±0,38 |
| FAL | $49,2^a$±0,28 | $45,8^b$±1,74 | $48,9^a$±0,97 | $48,2^a$±0,79 | $51,2^a$±0,30 | $50,0^a$±1,18 | $40,1^b$±0,38 | $50,4^a$±0,56 |
| EL | $19,8^a$±0,21 | $17,4^b$±0,60 | $21,0^c$±0,54 | $21,9^c$±0,62 | $21,9^c$±0,27 | $20,6^c$±0,52 | $15,9^b$±0,18 | $21,7^c$±0,29 |
| HL | $34,6^b$±0,82 | $28,2^{bc}$±3,77 | $52,8^a$±3,03 | $16,3^c$±5,07 | $42,3^{ab}$±1,06 | $40,8^{ab}$±4,65 | $16,8^c$±0,52 | $53,0^a$±1,79 |
| HG | $155,4^{cd}$±0,79 | $155,9^{cd}$±5,04 | $160,2^{bc}$±1,33 | $180,1^a$±3,03 | $162,0^{bc}$±0,97 | $174,9^{ab}$±3,84 | $144,2^d$±0,74 | $174,7^{ab}$±1,99 |
| HC | $34,6^b$±0,31 | $38,7^b$±0,86 | $46,5^a$±1,23 | $47,9^a$±1,20 | $38,2^b$±0,28 | $48,9^a$±0,90 | $37,9^b$±0,31 | $46,4^a$±0,70 |
| TL | $101,8^{ab}$±1,25 | $81,7^c$±2,53 | $106,8^{ab}$±2,14 | $118,4^a$±3,56 | $107,7^{ab}$±1,57 | $110,4^{ab}$±2,49 | $93,4^{bc}$±0,85 | $109,9^{ab}$±1,52 |
| SPW | $34,8^{ab}$±0,33 | $29,2^b$±0,80 | $33,6^{ab}$±0,88 | $35,4^{ab}$±1,87 | $36,4^a$±0,43 | $35,1^{ab}$±0,98 | $29,0^b$±0,35 | $36,6^a$±0,63 |
| HW | $42,5^a$±0,35 | $41,0^{ab}$±2,07 | $42,9^a$±1,09 | $44,9^a$±1,56 | $43,7^a$±0,41 | $45,5^a$±0,91 | $36,4^b$±0,40 | $47,4^a$±0,55 |
| CD | $62,9^b$±0,40 | $56,6^b$±1,40 | $59,0^b$±0,61 | $65,4^{ab}$±1,22 | $64,7^{ab}$±0,48 | $63,6^{ab}$±1,40 | $51,8^b$±0,48 | $68,5^a$±0,90 |
| WH | $117,0^b$±0,58 | $109,2^{bc}$±1,22 | $137,0^a$±1,41 | $135,1^a$±1,41 | $128,8^{ab}$±0,52 | $130,4^a$±2,09 | $101,8^c$±0,51 | $136,5^a$±0,94 |
| RH | $123,6^{bc}$±0,49 | $111,6^c$±2,87 | $139,7^a$±1,89 | $139,0^a$±1,58 | $133,8^a$±0,59 | $134,3^a$±2,00 | $107,0^c$±0,57 | $138,2^a$±0,64 |
| BL | $119,4^b$±0,71 | $115,8^{bc}$±5,26 | $134,1^a$±2,26 | $132,1^a$±4,53 | $123,2^b$±0,94 | $124,4^b$±1,92 | $103,2^c$±0,67 | $131,8^a$±1,02 |
| SIL | $163.8^b$±1.12 | $151.4^c$±1.70 | $198.4^a$± 0.90 | $182.8^b$± 1.46 | $173.6^{ab}$±3.15 | $184.8^a$± 3.10 | $144.7^c$±1.70 | $186.8^a$±1.39 |

[abcd] Means with different letters in rows are significantly different between locations at P ≤ 0.001; SNK's multiple mean comparison test

MC: Muzzle circumference, HW: Head width, FAL: Face length, EL: Ear length, HL: Horn length, HG: Heart Girth, HC: Hock circumference, TL: Tail length, SPW: Shoulder point width, HW: Hip Width, CD: Chest depth, WH: Withers height, RH: Rump height, BL: Body length, SIL: Scapula-ischial length

which generally presented a single reddish brown coat color. Taurine and crossbreed presented variable coat colors. In addition, about half of the animals sampled as of the *Bargouji* breed had no hump, and the same observation was made for the *Keteeji* breed.

## Typology of cattle breeds

The canonical coefficients showing the contribution of each measured morphometric trait to the total variation are presented in S4 Table. The first two canonical variates together accounted for 86.26% of the total variation among breeds. The correlation between the cattle breeds and CAN1 was 0.896 and the one between the breeds and CAN2 was 0.771 and the two axes were significant (P<0.001) and sufficient to classify all individual cattle studied. The variables height at withers (WH), rump height (RH), scapulo-ischial length (SIL), horn length (HL), body length (BL), hip width (HW), heart girth (HG), chest depth (CD), ear length (EL), and hock circumference (HC) proved to be the most useful in discriminating among the nine cattle breeds investigated.

The plot of the centroid values of these first two canonical discriminant functions (Fig 2) shows a clear cut separation between zebu and taurine breeds. The taurine breeds of *Boboji* and Somba appeared to be the two most homogeneous groups whereas *Bargouji* animals partially overlap with the *Keteeji*. Furthermore, there was no clear cut separation among zebu breeds and also between zebus and unnamed crossbreeds.

S5 Table shows the pairwise Mahalanobis distances among the nine cattle breeds investigated. All pairwise distances between the breeds were significant (P<0.001). The greatest distance was observed between Somba and *Bodeeji* (46.87), followed by Somba and *Dageeji* (33.89) whereas the smallest was observed between *Yakanaji* and *Bodeeji* (2.54) followed by *Yakanaji* and crossbreeds (2.59).

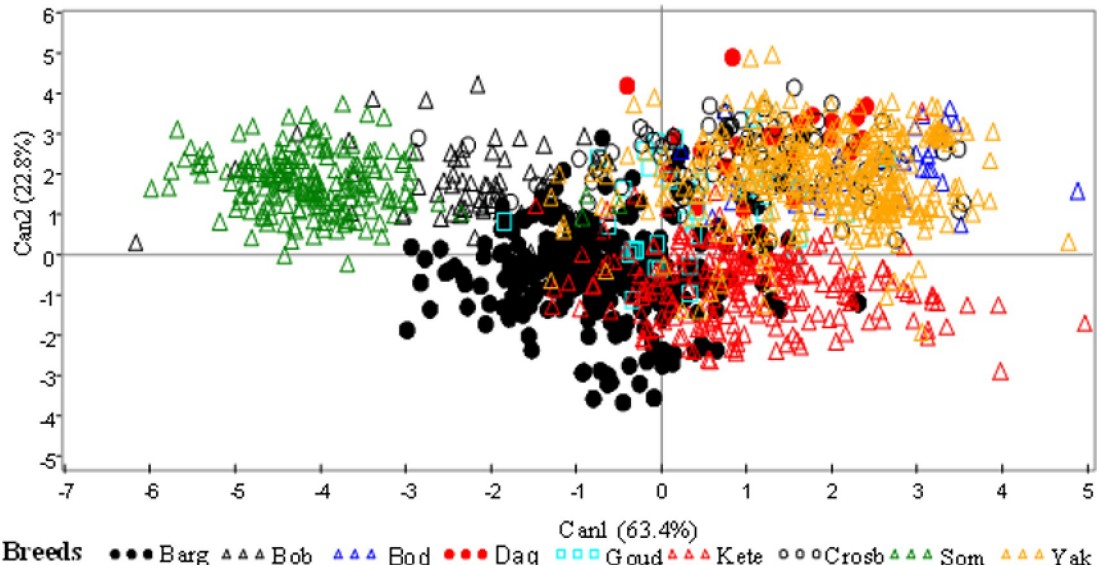

**Fig 2. Scatterplot of 1401 individual animals on the first two canonical discriminant functions.**

The dendrogram based on the distance matrix (Fig 3) shows three main clusters: Group 1 included the *Bargouji* breed which was clearly separated from zebu breeds but close to the two taurine breeds of *Somba* and *Boboji* (Cluster 2). The third cluster included all zebu breeds, the unnamed crossbreeds and the *Keteeji*.

The two discriminating functions correctly classified about 75% of the individuals into their a-priori groups (Table 6). However, it is worth noting that about half of the *Yakanaji* and

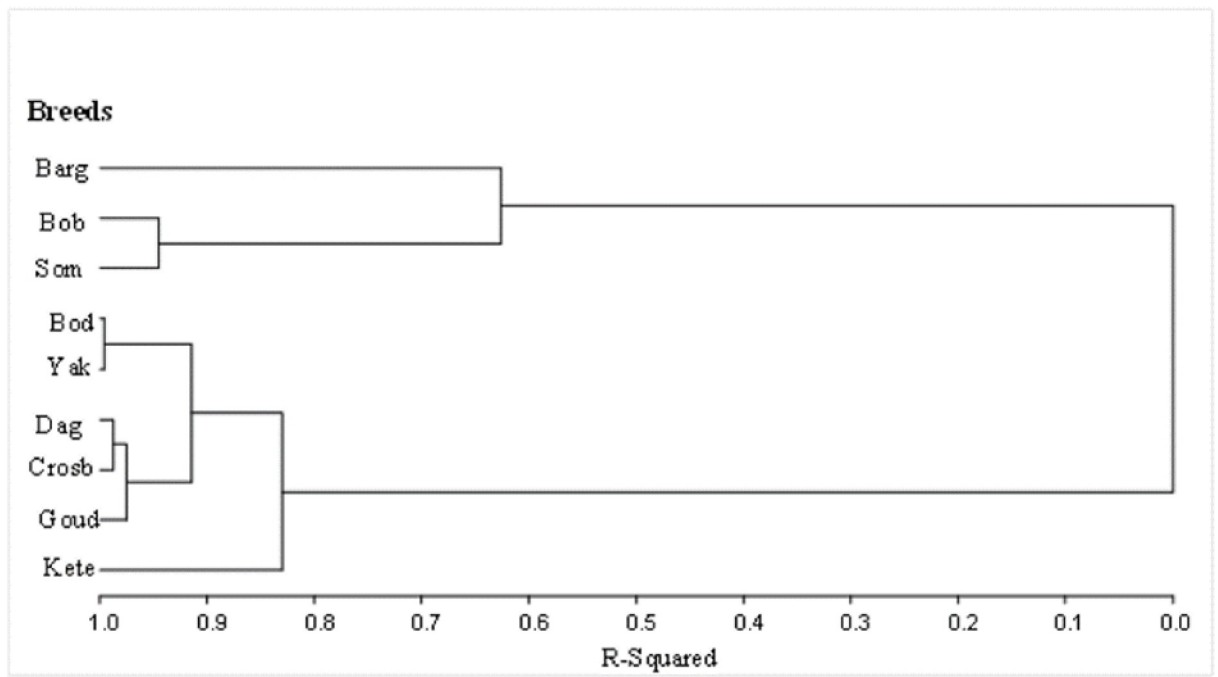

**Fig 3. Cluster analysis (UPGMA) of cattle breeds based on Mahalanobis distances.**

**Table 6. Percent (%) of individual cattle classified into their a-priori breeds.**

| Breed | Posterior probability (%) | | | | | | | | | Total |
|---|---|---|---|---|---|---|---|---|---|---|
| | *Bargouji* | *Boboji* | *Bodeeji* | *Dageeji* | *Goudali* | *Keteeji* | *Crossbreed* | *Somba* | *Yakanaji* | |
| *Bargouji* | **80.71** | 1.78 | 0.00 | 1.19 | 0.89 | 11.57 | 1.48 | 0.59 | 1.78 | 337 |
| *Boboji* | 0.00 | **84.13** | 0.00 | 0.00 | 0.00 | 0.00 | 4.76 | 11.11 | 0.00 | 63 |
| *Bodeeji* | 0.00 | 0.00 | **83.33** | 2.78 | 0.00 | 0.00 | 2.78 | 0.00 | 11.11 | 36 |
| *Dageeji* | 0.00 | 0.00 | 8.33 | **75.00** | 0.00 | 0.00 | 8.33 | 0.00 | 8.33 | 24 |
| *Goudali* | 0.00 | 0.00 | 0.00 | 0.00 | **84.38** | 0.00 | 6.25 | 0.00 | 9.38 | 32 |
| *Keteeji* | 9.96 | 0.43 | 1.30 | 0.00 | 0.00 | **83.55** | 2.60 | 0.00 | 2.16 | 231 |
| **Crossbreed** | 5.45 | 11.82 | 7.27 | 10.00 | 5.45 | 0.00 | **49.09** | 0.91 | 10.00 | 110 |
| **Somba** | 0.52 | 3.11 | 0.00 | 0.00 | 0.00 | 0.00 | 0.52 | **95.85** | 0.00 | 193 |
| *Yakanaji* | 4.27 | 1.33 | 14.13 | 15.20 | 2.67 | 4.80 | 20.00 | 0.00 | **37.60** | 375 |
| **Rate** | 0.192 | 0.158 | 0.166 | 0.250 | 0.156 | 0.164 | 0.509 | 0.041 | 0.624 | 0.251 |
| **Priors** | 0.111 | 0.111 | 0.111 | 0.111 | 0.111 | 0.111 | 0.111 | 0.111 | 0.111 | 0.111 |

**NB**: The percentage of well classified cow is read on the first diagonal (in bold)

crossbreed individuals were incorrectly classified whereas almost all Somba animals were successfully assigned to their original group.

## Correspondence analysis of breeds' qualitative traits

The Multiple Correspondence Analysis (MCA) of the qualitative characters produced slightly different results from those obtained with the Canonical Discriminant Analysis of the quantitative traits (Fig 4). The first dimension separated the taurine breeds (Group I) and the group of *Bodeeji* breed (Group II), from a very heterogeneous group (Group III) which included the other zebu breeds, the unnamed crossbreeds, the *Keteeji* and the *Bargouji* breeds. Group I was characterized by spotted and composed aspects of their coat with dominant black, white or brown colors, cup and numeral three forms of horns and absence of hump whereas Group II was associated with a reddish or red spotted white coat color. The animals of the third group, which is the most heterogeneous, had a dominant white coat color.

## Discussion

The originality and the innovative character of our approach in the West African extensive livestock production systems lie in linking herders' description of cattle breeds with metric measures of phenotypes. Herders' accuracy in describing and distinguishing different cattle breeds underlines the relevance of this approach for investigating the phenotypic and genetic variability in farm animal genetic resources. As observed in this study, herders' classification offers the advantage to be easily comparable to the quantitative classification obtained from the metric measures of phenotypes, as the discriminating criteria used by herders are in agreement with those most commonly used in morphological characterization studies in livestock [25].

There was also a considerable consistency in the local names attributed to each of the identified cattle breeds regardless of the geographic location of the herders. Cattle breeds were mainly named in *Fulfulde* language, the language of *Fulani/Fulbe* people, as the majority of pastoralists in West Africa belong to this ethnic group [44]. The breeds names given by cattle herders have previously been reported by many authors and match with the majority of cattle breeds commonly found in West Africa: *Yakanaji* known as *Daneeji/Akuji/Bunaji/*White Fulani [15,44], *Bokoloji/Goudali* or *Zomanta* in *Fongbe* language (the most common native

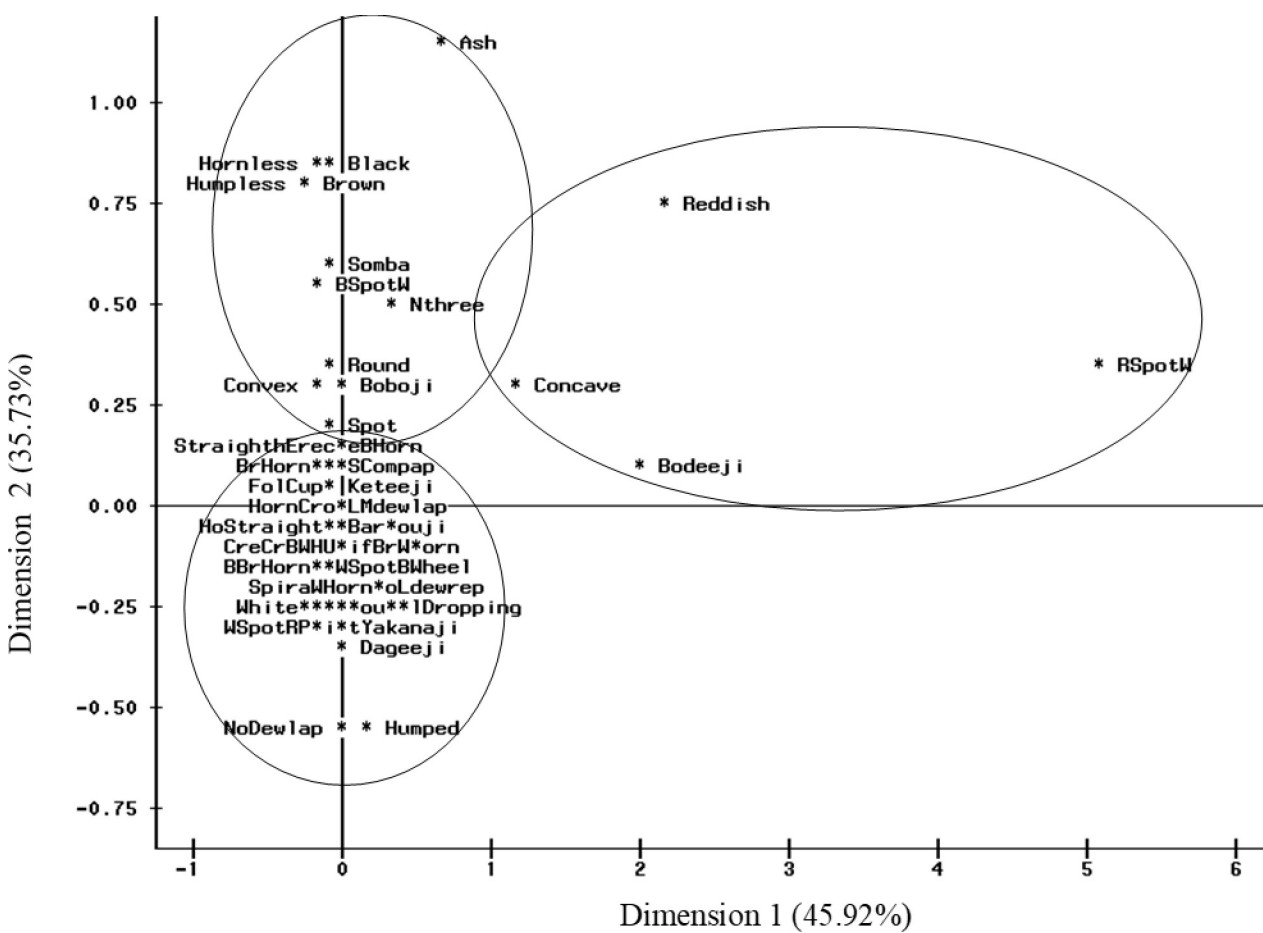

**Fig 4. Multiple correspondence analysis of the morphological traits of cattle breeds.**

language spoken in Benin) and known as Sokoto Goudali [15,44], *Bodeeji/Bororo* also called *WoDaaBe*, Red Fulani or Red Bororo [44,45], *Djelliji/Djelli/Djalli* known as Peulh Nigérien [46], *Dageeji* also called *Dage* [47], and *Bargouji/Bargou* commonly designated as Borgou [48]. Somba and *Keteeji* are also reported in the literature under these same names [48, 49,50,51]. *Boboji* was the only cattle name that does not appear in the existing literature. According to its traits, as described by the herders, it is a cattle of small size, humpless with shorthorns, and reported to be mainly found in the southern regions of Benin. The given characteristics are most similar to those of the shorthorn lagoon taurine cattle, commonly known as Lagunaire or Lagune [52].

The findings of this study reveal the diversity of cattle breeds traditionally raised in Benin and confirm that herders have a good knowledge of their animal genetic resources [53]. The number of breeds reported by the surveyed herders in this study is however greater than those reported in previous studies. For instance, a recent study conducted in northern Benin [53] reported only five cattle breeds (*Keteeji*, *Jaliji/Djelliji*, *Bodeeji*, *Tchiwali/Yakanaji* and Goudali) out of the total of ten recorded in the current study. However, the last authors' study differs with respect to geographical focus. Indeed, in contrast to our study, which considered seven localities along the geographical north-south gradient in Benin, the investigation by [53] was limited to the Biosphere Reserve of W National Park, in the extreme north of the country. Likewise, the national report to FAO in 2004 [54] also mentioned the presence of eight distinct

cattle breeds, omitting the *Dageeji* cattle, which has never been mentioned in previous studies. Main reasons for its neglect may include its small population size and limitation to a few herds in regions not considered in these studies as pastoral areas. Similarly, the Domestic Animal Genetic Resources Information System (DAGRIS), the web-based electronic source of information on selected indigenous farm animal genetic resource [55] reports fewer cattle breeds for Benin than the current study.

Herders' description of cattle breeds clearly differentiated the humpless small-framed *Boboji* and Somba cattle from the humped and larger-framed zebus. This difference between zebu and taurine cattle subspecies was confirmed by the multivariate analysis performed on the recorded metric traits. Similar results have been obtained through other studies [16,56]. In the same way, the closeness of the *Boboji* and Somba breeds to each other in their physical appearance as mentioned by our respondents have been confirmed by the classical methodologies (AHC and MCA) of classifying farm animal genetic resources based on measured morphometric traits. However, the scatterplot displaying all individual animals on the canonical discriminant functions successfully separated these two breeds, with almost no overlap. This finding is consistent with the results from a previous molecular comparison of the two breeds by [56] and suggests that the morphological characterization may be appropriate and sufficient to study and compare the genetic structure in these two morphologically close breeds.

In contrast, with the exception of the *Bodeeji*, which was separated from others zebu in MCA performed with qualitative traits, the zebu breeds distinguished by herders could not be accurately separated in the multivariate analyses. This underlines the relevance of qualitative information in the morphological description of livestock. Although often considered subjective [16, 57], well-provided information on the qualitative traits of animals could be useful for supplementing quantitative data and may also help in rapidly sampling an animal breed for in-depth studies such as molecular analyses.

The difficulty for a perfect separation of zebu cattle types corroborates the challenges in identifying genetic variation patterns among West African livestock breeds from their morphology as previously reported [17,58]. These authors explain this situation by the lack of selection and high levels of gene flow due to cyclical cross-border cattle herd movements known as "transhumance" and to extensive commercial transactions of cattle on the hoof in the West African region. For these zebu breeds, further discrimination at the molecular level may be necessary.

The Mahalanobis distance obtained between breeds showed, however, a significant variation in the morphological closeness among breeds. The low values of Mahalanobis distances between *Yakanaji* and several other zebu breeds, for instance *Bodeeji* and *Dageeji*, as well as between *Yakanaji* and unnamed crossbreeds reveal a high degree of overlap in morphological characteristics among these breeds. *Yakanaji*, in fact, appeared to be the most heterogeneous zebu breed in the study area as revealed by the low percentage of individuals from this breed correctly classified in their a-priori group. The aforementioned overlap hampers the differentiation of these breeds on the exclusive basis of multivariate analyses of their morphometric traits.

The great heterogeneity of the *Yakanaji* cattle might result from their large use in crossbreeding by livestock herders [52] due to their good production performance and adaptive traits [44,59,60]. Unsupervised and indiscriminate crossbreeding with local cattle breeds, as often encountered, represents an important threat to the conservation and sustainable use of the latter.

In contrast to *Yakanaji*, the highest values of Mahalanobis distances between Goudali and the other zebu cattle breeds are consistent with the specific characteristics of this cattle breed especially its conformation and the absence of horn. Further, this breed is less used in

crossbreeding in the surveyed areas. Its sensibility to trypanosomiasis [61,62], an infectious disease caused by a protozoan parasite, could explain its geographical restriction in Benin to the virtually tsetse free Northern Sudanian zone.

Interestingly, there has been a considerable inconsistency in the classification of *Bargouji* and *Keteeji* cattle breeds using multivariate techniques. While the AHC performed on the quantitative traits clearly separated the two cattle types, they were grouped together in the MCA approach using their qualitative traits. In the scatterplot (Fig 2) as well as in the dendogram (Fig 3), individuals from the *Bargouji* breed were very close to the taurine breeds (*Somba and Boboji*) in their morphological characteristics whereas *Keteeji* were grouped with zebus and unnamed crossbred. This important overlap between zebus and *Keteeji* is certainly due to the presence of hump in more than half of the individuals recorded as *Keteeji*. But herders considered the "true" *Bargouji* as a humpless animal; they also firmly identified some humped individuals as *Bargouji* and the "true" *Keteeji* as a humped animal. We can therefore argue that there exist two sub-types (one humped and one humpless) in each of the two cattle breeds. These results, in contrast to those obtained with the analysis of the qualitative traits, are in congruence with herders' classification who exhibited a very good knowledge of the two cattle breeds. It highlights that neither qualitative nor quantitative traits alone are sufficient in breed characterization, but a good combination of both.

But, both cattle breeds of *Bargouji* and *Keteeji* show body size values intermediate between those of taurine and zebu subspecies which certainly explains the difficulty to distinguish them in previous scientific reports. [48] recognized the existence of both the *Keteeji* and the *Bargouji* (*Barguuji*/ Borgou according to the author) in Benin but considered the latter as a taurine and the *Keteeji* as a crossbreed between zebu and taurine. According to [51], who reported its presence since 1918 in the Niger Valley in the Northern Sudanian zone of the country, the *Keteeji* is a "crossbreed between the zebu and the small *N'dama* cattle from the more humid south Borgou". Yet, the *N'dama* breed, a longhorn taurine cattle native of Guinea, was introduced to Benin only in 1952 [63]. The absence of *N'dama* cattle among the cattle breeds elicited by the herders participating in the present study is consistent with the progressive disappearance of this breed in Benin, previously highlighted [54].

Also, the recent study [53] confounded *Keteeji* to *Borgu* (Borgou). In the official reports on cattle breeds from Benin and West Africa, these two cattle breeds are indiscriminately referred to as "Borgou" cattle, a "stabilized" crossbreed between *Yakanaji* and Somba [64], even though more than half of the Borgou herds were further mated with zebus [65]. One of the issues that emerge from these findings and that has already been pointed out [7,52] is the heterogeneity in many "stabilized" crossbreeds in West Africa. Hence, our results provide support for the hypothesis that *Keteeji* and *Bargouji* are two varieties of the Borgou cattle. It also confirm the necessity of combining molecular analyses, phenotypic characterization and herders' knowledge for a more accurate differentiation of the breeds and subtypes of cattle raised in extensive African livestock production systems for their effective management and preservation. Several of them have already disappeared before being formally identified [12].

## Conclusions

The aim of this study was to validate pastoralists' classification system and knowledge of cattle breeds with quantitative morphometric analyses. The results showed the need to associate qualitative traits and quantitative traits measurement in morphological discrimination of cattle breeds. The findings reveal that livestock herders have a good knowledge of the morphological trait characteristics of the cattle breeds raised in their herds. The multivariate analyses of the morphometric traits showed less accuracy than herders' classification approach in discriminating most of the

zebu breeds because of high variability within and among breeds. The difficulty of perfect separation of these cattle breeds, whatever the single approach of classification, suggests combining livestock herders' traditional knowledge with phenotypic and molecular genetic approaches as an integrated tool for the appropriate characterization of farm animal genetic resources.

## Ethics approval and consent to participate

The study involved taking body measurements from cattle with the consent and in the presence of the cattle herder. There is no specific legislation for body measurements and hence no approval was necessary. All the data was collected in traditional farms and the animal owners agreed to be involved in the project through the Communal Sector for Agricultural Development (SCDA), which is the decentralized institution for the management of the agricultural sector in the surveyed municipalities. All the animals included in this study were managed in accordance with the criteria for the assessment of animal welfare identified by the Welfare Quality Project (WQP) [42].

## Supporting information

**S1 File.**
(XLSX)

**S1 Table. Least square means in cm (± standard error) of morphological variables for nine cattle breeds raised in Benin (pooled data for both cows and bulls).**
(PDF)

**S2 Table. Distribution (in %) of measured qualitative traits among nine cattle breeds raised in Benin.**
(PDF)

**S3 Table. Distribution (in %) of measured qualitative traits among nine cattle breeds raised in Benin (continued).**
(PDF)

**S4 Table. Canonical loadings of fifteen measured morphological traits from nine cattle breeds raised in Benin on the first two canonical variables.**
(PDF)

**S5 Table. Pairwise Squared Mahalanobis distances among nine cattle breeds raised in Benin (based on data measured from both cows and bulls).**
(PDF)

**S1 Fig. Somba cow.**
(TIF)

**S2 Fig. *Boboji*/Lagunaire cow.**
(TIF)

**S3 Fig. *Yakanaji/Bunaji*/White Fulani cow.**
(TIF)

**S4 Fig. *Goudali/ Bokoloji* /Sokoto Gudali cow.**
(TIF)

**S5 Fig. *Bodeeji/ Bororo/* Red Bororo cow.**
(TIF)

**S6 Fig.** *Djelliji/Djalli*/**Peulh Nigérien bull.**
(TIF)

**S7 Fig.** *Dageeji/Dage* **cow.**
(TIF)

**S8 Fig.** *Bargouji / Keteeji*/**Borgou cow.**
(TIF)

## Acknowledgments

The authors express their sincere gratitude to all participating herders for their valuable help and cooperation during the field work.

## Author Contributions

**Conceptualization:** Sandrine O. Houessou, Luc Hippolyte Dossa, Rodrigue Vivien Cao Diogo, Eva Schlecht.

**Data curation:** Sandrine O. Houessou, Luc Hippolyte Dossa, Maurice Cossi Ahozonlin.

**Formal analysis:** Sandrine O. Houessou, Luc Hippolyte Dossa, Maurice Cossi Ahozonlin.

**Funding acquisition:** Luc Hippolyte Dossa.

**Investigation:** Sandrine O. Houessou.

**Methodology:** Sandrine O. Houessou, Luc Hippolyte Dossa, Mahamadou Dahouda.

**Writing – original draft:** Sandrine O. Houessou, Luc Hippolyte Dossa.

**Writing – review & editing:** Rodrigue Vivien Cao Diogo, Mahamadou Dahouda, Eva Schlecht.

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
