## [Decision Letter · Decision Letter 0]

11 Jul 2019

PONE-D-19-14824

Confronting pastoralists’ knowledge of cattle breeds raised in the extensive production systems of Benin with multivariate analyses of morphological traits

PLOS ONE

Dear Dr Dossa,

Thank you for submitting your manuscript to PLOS ONE. After careful consideration, we feel that it has merit but does not fully meet PLOS ONE’s publication criteria as it currently stands. Therefore, we invite you to submit a revised version of the manuscript that addresses the points raised during the review process.

We would appreciate receiving your revised manuscript by Aug 25 2019 11:59PM. To enhance the reproducibility of your results, we recommend that if applicable you deposit your laboratory protocols in protocols.io, where a protocol can be assigned its own identifier (DOI) such that it can be cited independently in the future. For instructions see: http://journals.plos.org/plosone/s/submission-guidelines#loc-laboratory-protocols

We look forward to receiving your revised manuscript.

Kind regards,

Juan J Loor

Academic Editor

PLOS ONE

Journal Requirements:

http://www.ccsenet.org/journal/index.php/sar/article/view/0/39141

In your revision ensure you cite all your sources (including your own works), and quote or rephrase any duplicated text outside the methods section. Further consideration is dependent on these concerns being addressed.

Reviewers' comments:

Reviewer's Responses to Questions

**Comments to the Author**

1. Is the manuscript technically sound, and do the data support the conclusions?

Reviewer #1: Yes

Reviewer #2: Yes

Reviewer #3: Yes

2. Has the statistical analysis been performed appropriately and rigorously? 

Reviewer #1: Yes

Reviewer #2: Yes

Reviewer #3: Yes

3. Have the authors made all data underlying the findings in their manuscript fully available?

Reviewer #1: Yes

Reviewer #2: Yes

Reviewer #3: Yes

4. Is the manuscript presented in an intelligible fashion and written in standard English?

Reviewer #1: Yes

Reviewer #2: Yes

Reviewer #3: Yes

5. Review Comments to the Author

Reviewer #1: The paper Confronting pastoralists’ knowledge of cattle breeds raised in the extensive production systems of Benin with multivariate analyses of morphological traits is a good paper.

In my opinion need:

1. In Table 1:) Suggest drop this information [Region; Agroecological zone (a); Soil type (a); Rainy season] on the table. Write in Material and Method (text body)

2. In Line 58: Version of SAS (2008) – is a paid program. Need use a actual version or use a SAS Student; (49. SAS (Statistical Analysis System Software), 2008. SAS Version 9.2, SAS Institute Inc, 512 Cary, NC, USA.)

3. The authors can insert an image et one cattle bread (Supporting information); Need insert the general names of each breed if had.

4. On the table 3: what’s means Trait expression ;;) coat color Variable?

Reviewer #2: This paper is valuable in that it helps to fill the large data-gap on African cattle. The paper is also well written and statistically sound.

My main comments are to (a) reduce the length of the introduction, (b) add a disccussion on the meaning of 'breed' which is subjective, and how you define it in this work.

If you do have tissue / blood / hair samples from the animals you measured, it would be extremely interesting to compare genomic results to those presented here.

Reviewer #3: Comments to the Author

Comments about the manuscript entitled “Confronting pastoralists’ knowledge of cattle breeds raised in the extensive production systems of Benin with multivariate analyses of morphological traits”.

Authors investigated on the pastoralists’ knowledge about of the cattle breeds raised in Benin. Methods used were clearly explained, but some precisions are necessary. Obtained results must be encouraged.

In general, this manuscript can be accepted for publication. Further remarks are given below.

Abstract

L21: change “characteristics-” to “characteristics.”

L34: Change ‘‘keepers’’ to ‘‘herders’’ and harmonize it in all the text

Introduction

L56-58: Add reference sources

L91-95: Change ‘‘[42] further argues that in traditional livestock production systems, …. phenotypic traits.’’ to ‘‘Further, in traditional livestock production systems, …. phenotypic traits [42].’’

L97-98: Move ‘‘[47,48]’’ at the end of the sentence

L99: Change ‘‘with the aim’’ to ‘‘in order to’’

L100-102: Rephrase to show clearly the study aim like that is in the abstract

Material and methods

How were interviewed herders sample population fixed? The study may justify that

Table 1: Change ‘‘(a)’’ to [70]; ‘‘(b)’’ to [72] and ‘‘(c)’’ to [71]

Table 1: Change ‘‘Estimated total cattle population’’ to ‘‘Estimating of the total cattle population’’

L1: Harmonize expression ‘‘cattle farmer’’ or ‘‘cattle herder’’ in the all of the text

L6-7: CARDER was the departmental institution. So Communal Sector of the Agricultural Development (SCDA) was the local institution in the municipalities.

L11: Delete ‘‘reported by’’

L17: Rephrase ‘‘identify the villages’’

L59-60: Why this exclusion? Were these 6 animals included in the 1401 individual cattle?

Results

L88-89: What were the language of these breed names?

L97-99: What about of Keteeji Bargouji ? are they taurine or zebu breeds?

L105: Table 3: 8 breeds were described. What about of ‘‘Keteeji’’ ?

L105: Table 3: Goudali or Gudali? Harmonize it

Discussion

L207: Write ‘‘Fongbe language’’

L219-221: Rephrase: ‘‘For instance, a recent study conducted in ??? reported only five cattle breeds (Keteeji, Jaliji/Djelliji, Bodeeji, Tchiwali/Yakanaji and Gudali) [48] out of the total of ten recorded in the current study’’

L221: Change ‘‘their study’’ to ‘‘the last authors study’’

L234: Rephrase ‘‘by [16,60]’’ to ‘‘through others studies [16,60]”

L252: Rephrase ‘‘as previously reported by [17,62]’’ to ‘‘as previously reported [17,62]’’

Ovoid ‘‘by […]’’ and harmonize it in the all of the text.

6. PLOS authors have the option to publish the peer review history of their article (what does this mean?). If published, this will include your full peer review and any attached files.

Reviewer #1: No

Reviewer #2: No

Reviewer #3: Yes: Sèmanou Robert DOGNON

---

## [Author Response · Author response to Decision Letter 0]

25 Aug 2019

Please see attached file named "Response to Reviewers"

Academic Editor (Comments provided as attached pdf file)

Comment 1: Recent analysis has shown there are no pure Bos Indicus in Africa - delete and just keep the term Zebu

Response: We thank the editor for this comment. We Have delete “Bos indicus” as suggested. Line 52

Comment 2: Rephrase 'they are mostly characterised by various adaptive and performance traits' as it doesn't add much meaning. E.g. to 'adaptive traits they are characterised by inlcude......'

Response: The sentence has been revised as follows: They are appreciated for their adaptive traits which include the resistance to diseases and drought, ability to walk long distances, and capacity to survive on poor pastures [2]. Line 54-56

Comment 3: I don't agree with high fertility - reproductive performance of African cattle is typically very low, often because of the poor management conditions / harsh environment. This should be removed.

Response: This was reported in the review paper by Mwai et al 2015. But as suggested by the reviewer, we have removed “high fertility”. Line 54-56

Comment 4: This sentence “Indeed, the characterization of livestock breeds can be performed through well-designed phenotypic, genetic and molecular studies that include pertinent and well thought-out analysis and interpretation of quantitative data.” seems to contradict the previous where you suggest that only doing phenotypic, genotypic and molecular characterisation is not sufficient. .....I think it is a grammer problem (more than the point you are trying to make)

Response: We thank the editor for this remark. While shortening the introduction as suggested by most reviewers, this sentence has been removed.

Comment 5: Yes and also all the other phenotypic chacterisations that are beyond morphological e.g. calving interal, milk yield, disease resistance and so forth

Response: We thank the editor for this important detail.

Comment 6: Here I would suggest changing no studies to limited studies, or 'no studies of which we are aware'

Response: We thank the editor for this comment. Changes have been made as follows:

So far, however, no studies of which we are aware combine the two approaches in order to test their complementarity and validate pastoralists’ classification of cattle breeds. Line 93-95

Comment 7: The length of the introduction should be reduced by at-least one-third. Also here, or in the methodology, the concept of breed needs to be explained and the definition of breed you are using in this article given (see e.g. SoW AnGR)

Response: We thank the editor for the comments. 

We have reduced the length of introduction section by about one-third of the initial characters. 

 A discussion has also been provided in Material and methods section on the meaning of “breed” as suggested by the editor as follows:

 Breed was defined according to [43] as: “either a sub-specific group of domestic livestock with definable and identifiable external characteristics that enable it to be separated by visual appraisal from other similarly defined groups within the same species, or a group for which geographical and/or cultural separation from phenotypically similar groups has led to acceptance of its separate identity”. Thus, cattle breed in this study is referred to cattle of similar physical features perceived by herders as being of the same genetic origin. [Line 137-142]

Comment 8: This paragraph can be consdierably shortened - you don't need to mention the things your study did not do (adverse animal handling etc.)

Response: Following the reviewer’s suggestion the paragraph has been shortened. [Line 119-126]

Comment 9: This paragraph can be said more briefly - all these details are not reqwuired (community sensitisation is standard practice before any research engagement) 

Again remove unncecessary details e.g. the statement about arranging a suitable time 

Add dates of the study

Response: 

Following the reviewer’s suggestion some details have been removed. But, other important information as requested by the reviewers (dates of the study, the way of herders’ population sampling selection, the discussion about the meaning of “breed”) have been kept in the paragraph.

Comment 10: be consistent on use of 'or' vs '/'

Response: We thank the editor for this observation. Changes have been made accordingly.

Reviewer #1: 

General comment: The paper Confronting pastoralists’ knowledge of cattle breeds raised in the extensive production systems of Benin with multivariate analyses of morphological traits is a good paper.

Response: We thank the reviewer for his encouraging comments on the manuscript.

In my opinion need:

1. In Table 1:) Suggest drop this information [Region; Agroecological zone (a); Soil type (a); Rainy season] on the table. Write in Material and Method (text body)

Response: We have revised the Table1 and added the information in the text body as suggested by reviewer [Lines 101-108]

2. In Line 58: Version of SAS (2008) – is a paid program. Need use a actual version or use a SAS Student; (49. SAS (Statistical Analysis System Software), 2008. SAS Version 9.2, SAS Institute Inc, 512 Cary, NC, USA.)

Response: The citation has been revised accordingly [Line 167-168]

3. The authors can insert an image et one cattle bread (Supporting information); Need insert the general names of each breed if had.

Response: Animal breed images have been added in supporting information as suggested.

4. On the table 3: what’s means Trait expression ;;) coat color Variable?

Response: Variable named “Coat color” referred to the “Body hair coat color” as suggested in a scientific reference paper [Food and Agriculture Organization. (2012). Phenotypic characterization of animal genetic resources.]

Reviewer #2: 

This paper is valuable in that it helps to fill the large data-gap on African cattle. The paper is also well written and statistically sound.

Response: We thank the reviewer for his encouraging comment.

My main comments are to (a) reduce the length of the introduction, (b) add a disccussion on the meaning of 'breed' which is subjective, and how you define it in this work.

Response:

a) We have reduced the length of introduction section by about one-third.

b) A discussion has been provided on the meaning of “breed” as suggested by reviewer as follow:

 Breed was defined according to [43] as: “either a sub-specific group of domestic livestock with definable and identifiable external characteristics that enable it to be separated by visual appraisal from other similarly defined groups within the same species, or a group for which geographical and/or cultural separation from phenotypically similar groups has led to acceptance of its separate identity”. Thus, cattle breed in this study is referred to cattle of similar physical features identify by herders as being of the same genetic origin.

[lines 137-142]

If you do have tissue / blood / hair samples from the animals you measured, it would be extremely interesting to compare genomic results to those presented here.

Response: We confirm that hair samples have been collected and cattle breeds presented in the manuscript are being compared on the genomic scale in a subsequent study.

Reviewer #3: 

Comments about the manuscript entitled “Confronting pastoralists’ knowledge of cattle breeds raised in the extensive production systems of Benin with multivariate analyses of morphological traits”.

Authors investigated on the pastoralists’ knowledge about of the cattle breeds raised in Benin. Methods used were clearly explained, but some precisions are necessary. Obtained results must be encouraged.

In general, this manuscript can be accepted for publication. Further remarks are given below.

Response: We thank the reviewer for his comments and observations; We have revised the manuscript accordingly.

Abstract

L21: change “characteristics-” to “characteristics.” 

Response: We thank the reviewer for his observation. Changes have been made accordingly. [L33]

L34: Change ‘‘keepers’’ to ‘‘herders’’ and harmonize it in all the text

Response: We thank the reviewer for his observation. Changes have been made accordingly throughout the document

Introduction

L56-58: Add reference sources

Response: Reference has been added. [L68]

L91-95: Change ‘‘[42] further argues that in traditional livestock production systems, …. phenotypic traits.’’ to ‘‘Further, in traditional livestock production systems, …. phenotypic traits [42].’’

Response: This sentence has been removed while reducing the introduction length as suggested by another reviewer.

L97-98: Move ‘‘[47,48]’’ at the end of the sentence

Response: This sentence has also been removed in order to reduce the introduction length as suggested by another reviewer. 

L99: Change ‘‘with the aim’’ to ‘‘in order to’’

Response: Change made and the sentence revised accordingly [L94]

L100-102: Rephrase to show clearly the study aim like that is in the abstract

Response: The study aim has been rephrased as suggested by reviewer as follows: 

This study aimed to document and validate herders' knowledge of differences among cattle breeds raised in Benin with quantitative data. [line 95-96]

Material and methods

How were interviewed herders sample population fixed? The study may justify that

Response: The sample population was fixed as follows:

Sampling herds were chosen among those hold by herders who have at least 10 years of experience in cattle herding and supposed as resident herders (herders that reside in the municipality for 05 years at least).

We have now provided this information in the text by revising the sentence on Lines 132-134 as follows “Individual interviews were then conducted from June 2015 to June 2016 with 803 resident cattle herders randomly selected from those who had at least 10 years of experience in cattle herding and had spent at least five (05) years in the locality”.

Table 1: Change ‘‘(a)’’ to [70]; ‘‘(b)’’ to [72] and ‘‘(c)’’ to [71]

Table 1: Change ‘‘Estimated total cattle population’’ to ‘‘Estimating of the total cattle population’’

Response: Changes have been made in Table 1 as suggested.

L1: Harmonize expression ‘‘cattle farmer’’ or ‘‘cattle herder’’ in the all of the text

Response: We thank the reviewer for this observation. We have made changes throughout the document as suggested.

L6-7: CARDER was the departmental institution. So Communal Sector of the Agricultural Development (SCDA) was the local institution in the municipalities.

Response: Thanks again for the remark. Change has been made as suggested. [122-124]

L11: Delete ‘‘reported by’’

Response: Change has been made as suggested.

L17: Rephrase ‘‘identify the villages’’

Response: The expression has been removed as suggested by another reviewer.

L59-60: Why this exclusion? Were these 6 animals included in the 1401 individual cattle?

Response: We thank the reviewer for this observation. We have removed the six animals given the too small size of sampled individual measured in each of these breeds (five individual cattle of Djelliji breed and only one male of Dageeji breed), which could further compromise statistical comparison between breed. 

No, the 6 excluded animals were not included in the 1401 individual cattle.

Results

L88-89: What were the language of these breed names?

Response: The language of the breed name was “Fulfulde language”. This has been mentioned in results [Line 200] and in discussion section [Line 320]

L97-99: What about of Keteeji Bargouji ? are they taurine or zebu breeds?

Response: Keteeji and Bargouji are crossbreed animals between taurine and zebu breed. They may be referred to Borgou cattle but with a difference in zebu blood level. This difference has been discussed in the discussion section [Lines 396-418] 

L105: Table 3: 8 breeds were described. What about of ‘‘Keteeji’’ ?

Response: We thank the reviewer for this remark. Using these key morphological traits, herders did not sharply distinguish Bargouji from Keteeji, except for the larger in size than Bargouji. This has been mentioned in result section Lines 223-230 and discussed in discussion section Lines 396-418

L105: Table 3: Goudali or Gudali? Harmonize it

Response: We have retained “Goudali” and harmonize throughout the text.

Discussion

L207: Write ‘‘Fongbe language’’

Response: Change has been made as suggested. Line 324

L219-221: Rephrase: ‘‘For instance, a recent study conducted in ??? reported only five cattle breeds (Keteeji, Jaliji/Djelliji, Bodeeji, Tchiwali/Yakanaji and Gudali) [48] out of the total of ten recorded in the current study’’

Response: The sentence has been revised and study area has been added as follows: For instance, a recent study conducted in northern Benin [53] reported only five cattle breeds (Keteeji, Jaliji/Djelliji, Bodeeji, Tchiwali/Yakanaji and Gudali) out of the total of ten recorded in the current study. [Lines 336-338]

L221: Change ‘‘their study’’ to ‘‘the last authors study’’

Response: Change has been made as suggested. Line 338

L234: Rephrase ‘‘by [16,60]’’ to ‘‘through others studies [16,60]”

Response: Sentence has been rephrased as suggested

L252: Rephrase ‘‘as previously reported by [17,62]’’ to ‘‘as previously reported [17,62]’’

Ovoid ‘‘by […]’’ and harmonize it in the all of the text.

Response: Sentence has been rephrased as suggested. Line 371

And “by” has also been avoided accordingly in all the text.

---

## [Decision Letter · Decision Letter 1]

9 Sep 2019

[EXSCINDED]

Confronting pastoralists’ knowledge of cattle breeds raised in the extensive production systems of Benin with multivariate analyses of morphological traits

PONE-D-19-14824R1

Dear Dr. Dossa,

We are pleased to inform you that your manuscript has been judged scientifically suitable for publication and will be formally accepted for publication once it complies with all outstanding technical requirements.

With kind regards,

Juan J Loor

Academic Editor

PLOS ONE

Additional Editor Comments (optional):

Reviewers' comments:

Reviewer's Responses to Questions

**Comments to the Author**

1. If the authors have adequately addressed your comments raised in a previous round of review and you feel that this manuscript is now acceptable for publication, you may indicate that here to bypass the “Comments to the Author” section, enter your conflict of interest statement in the “Confidential to Editor” section, and submit your "Accept" recommendation.

Reviewer #1: All comments have been addressed

Reviewer #3: All comments have been addressed

2. Is the manuscript technically sound, and do the data support the conclusions?

Reviewer #1: (No Response)

Reviewer #3: Yes

3. Has the statistical analysis been performed appropriately and rigorously? 

Reviewer #1: Yes

Reviewer #3: Yes

4. Have the authors made all data underlying the findings in their manuscript fully available?

Reviewer #1: Yes

Reviewer #3: Yes

5. Is the manuscript presented in an intelligible fashion and written in standard English?

Reviewer #1: Yes

Reviewer #3: Yes

6. Review Comments to the Author

Reviewer #1: The authors answered the suggestions and I believe the work is projected to be published.

As a reviewer, I don't see what else to add.

Post it.

Reviewer #3: Authors gave response about all of my comments. I recommend publication of this paper. My congratulations to the authors

7. PLOS authors have the option to publish the peer review history of their article (what does this mean?). If published, this will include your full peer review and any attached files.

Reviewer #1: No

Reviewer #3: Yes: Sèmanou Robert DOGNON

---

## [Editor Report · Acceptance letter]

13 Sep 2019

PONE-D-19-14824R1 

Confronting pastoralists’ knowledge of cattle breeds raised in the extensive production systems of Benin with multivariate analyses of morphological traits 

Dear Dr. Dossa:

I am pleased to inform you that your manuscript has been deemed suitable for publication in PLOS ONE. Congratulations! Your manuscript is now with our production department. 

With kind regards,

on behalf of

Dr. Juan J Loor 

Academic Editor

PLOS ONE